# PWM2Vec: An Efficient Embedding Approach for Viral Host Specification from Coronavirus Spike Sequences

**DOI:** 10.3390/biology11030418

**Published:** 2022-03-09

**Authors:** Sarwan Ali, Babatunde Bello, Prakash Chourasia, Ria Thazhe Punathil, Yijing Zhou, Murray Patterson

**Affiliations:** Department of Computer Science, Georgia State University, Atlanta, GA 30303, USA; sali85@student.gsu.edu (S.A.); bbello1@student.gsu.edu (B.B.); pchourasia1@student.gsu.edu (P.C.); rthazhepunathil1@student.gsu.edu (R.T.P.); yzhou43@student.gsu.edu (Y.Z.)

**Keywords:** coronavirus, host specification, COVID-19, *k*-mers, position weight matrix, classification, clustering

## Abstract

**Simple Summary:**

The family of coronaviruses comprises a diverse set of strains and variants which cause diseases from the common cold to COVID-19. Moreover, they infect a wide array of hosts from bats, camels, birds, to humans. Studying coronaviruses through the lens of host specificity provides a unique perspective to understanding the evolution, diversity and dynamics of this family. In particular, this can reveal groups of different hosts infected by similar strains, giving clues on strains which were more likely to have evolved to jump from one host to another. In this work, we frame host specificity as a classification task, in designing a very compact numerical representation of the spike sequences of different coronaviruses. Based on this numerical representation, classification methods are able to detect the target host with high accuracy. Such an approach can used to efficiently scale to large volumes of sequences, in order to unveil trends in the host specificity of different coronavirus strains.

**Abstract:**

The study of host specificity has important connections to the question about the origin of SARS-CoV-2 in humans which led to the COVID-19 pandemic—an important open question. There are speculations that bats are a possible origin. Likewise, there are many closely related (corona)viruses, such as SARS, which was found to be transmitted through civets. The study of the different hosts which can be potential carriers and transmitters of deadly viruses to humans is crucial to understanding, mitigating, and preventing current and future pandemics. In coronaviruses, the surface (S) protein, or spike protein, is important in determining host specificity, since it is the point of contact between the virus and the host cell membrane. In this paper, we classify the hosts of over five thousand coronaviruses from their spike protein sequences, segregating them into clusters of distinct hosts among birds, bats, camels, swine, humans, and weasels, to name a few. We propose a feature embedding based on the well-known position weight matrix (PWM), which we call PWM2Vec, and we use it to generate feature vectors from the spike protein sequences of these coronaviruses. While our embedding is inspired by the success of PWMs in biological applications, such as determining protein function and identifying transcription factor binding sites, we are the first (to the best of our knowledge) to use PWMs from viral sequences to generate fixed-length feature vector representations, and use them in the context of host classification. The results on real world data show that when using PWM2Vec, machine learning classifiers are able to perform comparably to the baseline models in terms of predictive performance and runtime—in some cases, the performance is better. We also measure the importance of different amino acids using information gain to show the amino acids which are important for predicting the host of a given coronavirus. Finally, we perform some statistical analyses on these results to show that our embedding is more compact than the embeddings of the baseline models.

## 1. Introduction

The coronavirus (COVID-19) pandemic is caused by the severe acute respiratory syndrome coronavirus 2 (SARS-CoV-2). The pandemic has put millions of people at risk in numerous countries worldwide and caused an unprecedented public health crisis [1]. Although the origin of COVID-19 (SARS-CoV-2) in humans is still unknown, there are many theories that it could have been transferred to humans from bats [2]. Likewise, several related coronaviruses (CoVs) have been transmitted from other animals, such as SARS (SARS-CoV) from civets (civets are closely related to cats [3]), and MERS (MERS-CoV) from dromedary camels [4]. SARS-CoV-2, and other CoVs, belong to the family coronaviridae (of order nidovirales [5]), which is a large family of diverse enveloped, positive-sense single-stranded genomic RNA (+ssRNA) viruses that can bring about respiratory diseases in humans and animals [6]. They are grouped into five genera, namely, alphacoronavirus, betacoronavirus, gammacoronavirus, alphaletovirus, and deltacoronavirus. They infect a range of hosts such as humans, palm civets, bats, dogs, and monkeys, among others [7]. The alphacoronaviruses and betacoronaviruses mostly infect mammals, and the gammacoronaviruses mostly infect birds. The deltacoronaviruses infect both birds and mammals [8].

SARS-CoV-2 is the seventh member of the coronavirus family known to affect humans, and the other six are severe acute respiratory syndrome-CoV (SARS-CoV), HCoVs-NL63, HCoVs-OC43, HCoVs-HKU1, HCoVs-229E, and middle east respiratory syndrome-CoV (MERS-CoV) [9]. SARS-CoV-2 is similar to SARS-CoV, which led to the SARS epidemic in 2003, causing more than 8400 cases and approximately 800 deaths [10]. Compared to the known SARS-CoV virus, the novel SARS-CoV-2 has a lower mortality rate but a higher human-to-human transmission rate. Similarly, SARS-CoV-2 can have an adverse impact on the human body. It is highly infectious and is a matter of significant concern, since it can not only damage the respiratory system, gastrointestinal system, heart, and central nervous system, but also may lead to multi-organ failure, and eventually, death [11,12].

Monitoring zoonotic diseases and host specificity are integral to understanding disease dynamics. Sixty percent of known infectious diseases in humans and 75% of all emerging diseases are zoonotic, as reported by the United Nations Environment Program (UNEP) and the International Livestock Research Institute (ILRI) [13]. The study of the COVID-19 pandemic is of great significance, not only because it can help healthcare institutions to cope with the ongoing epidemic, but also because it allows researchers to learn more fundamentals about the family coronaviridae, which can provide new knowledge for the prevention of potential pandemics in the future. CoVs are widespread among birds and mammals and can be causes of zoonoses. A zoonosis is defined as any disease or infection that is naturally transmissible from vertebrate animals to humans by the World Health Organization, and COVID-19 has been classified as a zoonotic disease [14]. One important step to learning about zoonoses and understanding the current pandemic better is finding out how human infections began for SARS-CoV-2. CoVs can lead to various diseases in domestic animals, including dogs, swine, chickens, and cats. Although the origin of COVID-19 in humans is still unknown, genetic analysis results show that it is highly possible that SARS-CoV-2 originates from bats and utilizes the pangolin as an intermediate host [15,16,17].

The CoVs have an envelope membrane that is associated with five structural proteins, namely, the surface (S) protein, or spike protein, haemagglutinin-esterase protein (HE), membrane protein (M), envelope protein (E), and the nucleocapsid protein (N) [18]; see Figure 1. The spike protein is responsible for the binding and fusion between the virus and the host cell receptors, and also the infected host cells and adjacent uninfected cells [19]. The spike protein is further subdivided into two subunits, S1 and S2. The S2 subunit is then again further subdivided into five domains, namely, the fusion peptide (FP), two heptad-repeat regions (HR1 and HR2), the transmembrane domain (TM), and the cytoplasmic tail (CT), all of which play a key role in mediating the viral cell membrane fusion and entry [20,21]. Hence, the spike proteins of different CoVs largely determine their ranges of host specificity. Changes in spike protein sequences are reportedly sufficient to change tissue and species tropism and viral virulence [7,22,23]. The S protein is a trimeric transmembrane protein with a protrusion, or spike, on the viral surface, which is the key for binding to and entry into host cells. It is composed of the receptor binding domain or S1 subunit and an S2 subunit that harbor sequences for viral fusion to the cell membrane [7,22,23]. Due to their importance, using the specificity of spike proteins offers an approach to classifying the potential hosts of CoVs.

A common way to classify and understand the dynamics of viruses is to construct a phylogenetic tree of evolution using the sequencing data of the virus [24,25]. After the COVID-19 pandemic breakout, databases such as GISAID [26] collected a large number of sequence data of SARS-CoV-2 from researchers and clinicians worldwide, which can be used for phylogenetic tree inference. Many methods have been developed and applied for constructing phylogenetic trees, including the most similar supertree algorithm (MSSA) method [27], the MRP method [28], and the approximate maximum likelihood (ML) supertree method [29,30]. Upon similar methods, the state-of-the-art Nextstrain [24] and IQTREE2 [25] have been developed. However, these methods of building a phylogenetic tree for CoVs require high computational complexity, and the vast volume of sequence data can cause a scalability issue for phylogeny-based approaches [31]. For example, Nextstrain [24] is able to construct trees on thousands of sequences, whereas IQTREE2 [25] is able to scale to tens of thousands of sequences. There are currently millions of sequences available on GISAID alone—clearly viable alternatives are necessary. Here we study machine learning clustering and classification as an alternative to phylogenetic tree building.

Some efforts have been made to study the coronavirus host data [32] by using the one-hot embedding (OHE) approach to get fixed length feature embeddings for the spike sequence. OHE provides good predictions, but it has drawbacks, such as the high dimensionality of the feature vectors produced. Furthermore, the columns of the OHE-based vector have a linear relationship, which means that one variable can be easily predicted using the other variables. This behavior can cause parallelism and multicollinearity (when multiple features are correlated with each other) in high dimensions. The authors of [33,34] used the coronavirus spike sequences to classify different variants of COVID-19 using *k*-mer-based frequency vectors. Researchers have performed clustering on the COVID-19 spike sequences using the same *k*-mer-based frequency vector generation approach [35,36]. Although their approaches are effective in terms of predictive performance, the dimensionality of the feature vector representation is still high, which can create a very well-known problem in machine learning: the curse of dimensionality. Moreover, for each *k*-mer, it is necessary to find the appropriate bin dedicated to a specific *k*-mer (“bin matching”) which can be expensive in terms of computational cost.

Another possible solution, which is what we propose here, is the use of the position weight matrix (PWM), sometimes also called a position-specific weight matrix (PSWM) or position-specific scoring matrix (PSSM) [37]. It is a good representation of motifs in biological sequences. A motif is a nucleotide or amino acid sequence pattern that is widespread and has or is conjectured to have some biological significance. The PWM applies entropy and relative entropy towards identifying transcription factor binding sites (TFBSs), for example. A PWM contains information about the frequencies of nucleotides for each position in the form of weights. These log-odds or log-probability weights are used for computing the binding affinity score.

The PWM can be used to distinguish the binding sites from the sequence, a well-known method for de novo motif sequence finding. If we do not know about a possible motif in a given sequence, there are methods such as expectation maximization (EM) and Gibbs sampling, which uses the PWM. Inspired by this application, we computed an absolute score from the PWM while scanning the sequence for “motifs” (here *k*-mers) using a sliding window (of size *k*, see Figure 2) and computed the absolute score. We can find relevant information on the motifs based on the score calculated from the PWM. The higher the score, the more relevant the *k*-mer is.

In this paper, we propose an approach called PWM2Vec, a basic implementation of the position weight matrix (PWM) to generate a feature vector representation of a coronavirus spike sequence. Given a spike sequence, we first extract *k*-mers. From the *k*-mers, we generate the PWM (see Figure 3). After that, we assign a score to each *k*-mer by using the PWM to design a feature embedding and apply machine learning methods such as classification and clustering in this feature embedding. While this is inspired by methods for finding motifs (e.g., TF-promoter binding sites), our goal is to obtain a numerical representation of these *k*-mers generated from each sequence.

Our contributions in this paper are as follows:We propose an approach to generate a fixed-length numerical representation of a spike sequence using a PWM. The generated feature vectors could be used as input to any machine learning algorithm for tasks such as classification and clustering.Our proposed feature embedding approach contains more compact information and gives better results than the baselines in terms of classification and clustering.Our feature vector contains fewer dimensions than *k*-mer and one-hot encoding-based feature vectors (≈20-fold fewer dimensions than one-hot encoding and ≈4-fold fewer dimensions than *k*-mer-based embedding), which improves the runtimes for classification and clustering algorithms.We performed statistical analysis on the data and show the importance of different positions of amino acids that play key roles in the classification of different hosts. We validated the compactness of our proposed embedding from an orthogonal point of view.

The rest of the paper is organized as follows: Section 2 contains the previous work related to sequence classification, in general, and coronavirus spike sequence classification in particular. Section 3 contains the details about our proposed alignment-free method for spike sequence classification. Section 4 contains the experimental setup, dataset collection, and dataset statistics details. The results for our proposed method are given in Section 5. Finally, we conclude our paper in Section 6.

## 2. Related Work

Several machine learning approaches based on *k*-mers have been proposed in the literature for classification and clustering tasks [33,35,38,39]. More specifically, there are many classical algorithms for sequence classification [40,41]. Although these methods have been proven to be useful in some studies, it is not clear if they can be used in the context of coronavirus data. Furthermore, another major problem with all those methods is the high computational complexity of the algorithms (because of the high dimensional representation of the data), which can result in higher runtimes for the underlying classification algorithms.

Position weight matrix (PWM)-based approaches have been successfully applied for diverse sequence analysis, motif predictions, and identification studies. Several popular software applications and Web servers have been built based on the implementations of PWMs, e.g., the PWMscan software package [42] and PSI-BLAST [43]. Furthermore, many other advanced algorithms have been implemented to optimize PWM-based techniques: examples include MEME (multiple EM for motif elicitation) [44], based on expectation maximization (EM), and the Gibbs Sampler [45] for de novo motif discovery, which uses Gibbs sampling algorithms [46,47]. The MEME EM algorithm basically finds an initial motif and repeatedly uses EM steps to improve motifs until the PWM values do not improve further [44,48]. Furthermore, the BaMM (Bayesian Markov model) algorithm was built based on the Markov algorithm to model correlations among nucleotides at other positions—since the PWM cannot, because the method assumes probabilities at different sites are independent of each other [47]. The PWM method continues to be applied and extended. Log2PWMs is a simple implementation of PWMs extended to enable conversion or reconstruction of a PWM representation from a sequence logo [49].

The PWM is also used for the binding specificity of a transcription factor (TF) [50]. It can be used to scan a sequence for the presence of DNA words, which are comparatively more similar to the PWM than to the background [51,52]. Authors in [53] evaluated the Bayesian network and a support vector machine (SVM) on four different TF binding site-based datasets, and analyzed their performances using PWM. Authors in [54] developed a tree-based PWM algorithm to simulate the interaction between TF and its binding sites accurately. A new di-nucleotide PWM approach is proposed in [55] that outperforms the conventional mono-nucleotide PWM method. Moreover, the research done in [56] proposes an improved position weight matrix (IPWM) method to recognize prokaryotic promoters based on an entropy measurement. Using hepatitis C virus (HCV) nucleotide sequences, the authors of [57] designed a global PWM for the genotypes of HCV genomes. Then, using the PWM, signatures were selected from the 5’ NCR, CORE, E1, and NS5B regions of the HCV genome. The predictive power of the selected signatures was then evaluated for predicting the most common HCV genotypes and subtypes.

Aside from DNA analysis, the PWM can also be applied to amino acid sequences. Authors in [58] developed an approach involving the position-specific scoring matrix (PSSM), another name for PWM, to predict protein–protein interactions between protein sequences. First, each protein is transformed into a PSSM, and then the PSSMs are adapted to detect distantly related proteins, the quaternary structural attributes of the proteins, and the proteins’ folding patterns. The research of in [59] proposes a PWM-based algorithm to predict signal peptide sequences and their cleavage positions in the amino acid sequences of bacteria and eukaryotes. Authors in [60] developed a PWM-based method for protein function prediction and proposed an argument for why the PWM and associated features have great potential for protein sequence analysis. Although the above methods are successful in their respective domains, they do not provide general means of designing a feature embedding for the underlying sequence, which contains rich information about the sequence and can be used as input to various machine learning algorithms.

The design of efficient feature vector-based representations has been studied in many domains, such as graph analytics [61,62], smart grid [63,64], electromyography (EMG) [65], clinical data analysis [66], network security [67], and text classification [68]. After the spread of COVID-19, efforts have been made to study the behavior of the virus using machine learning approaches. Several methods have been proposed recently for the classification of spike sequences. Authors in [33,69] used *k*-mers along with a kernel-based approach to classify SARS-COV-2 spike sequences. Authors in [32] proposed the use of one-hot encoding to classify the viral hosts of coronaviridae using spike sequences only. Although they were able to achieve higher predictive performance, some researchers in [33] showed that the *k*-mer-based approach performs better than the one-hot based approach, since it preserves sequence order information more effectively. Efficient clustering of spike sequences was performed in [35].

## 3. Proposed Approach

This section proposes an approach, PWM2Vec, to generate a fixed-length numerical feature embedding from coronavirus spike sequences for host specification. We also discuss the baseline approaches, specifically one-hot embedding (OHE) [32,34] and *k*-mer-based feature embedding [33,34]. We perform feature selection using ridge regression [70] on the resulting embedding before applying machine learning (ML) algorithms. This helps to reduce the dimensionality of the embedding, and hence the training time of the downstream ML algorithms.

### 3.1. One-Hot Embedding (OHE)

Machine learning algorithms require the input to be in a numerical format. It is necessary to process the (spike protein) sequence data into some numerical representation to apply these algorithms. One-hot embedding (OHE) [32] is a typical approach for obtaining a fixed-length numerical representation from sequence data. Considering an alphabet Σ, which contains the characters (amino acids) of the spike protein sequence, we need to map each character of Σ to a numerical (binary 0–1 vector) representation. We have 20 unique amino acids in the protein sequence data, namely, “ACDEFGHIKLMNPQRSTVWY.” We designed a feature vector for each amino acid. Each symbol has a length of 20 and has a value of 1 corresponding to the position of the character in the alphabet, and 0 for all other places in the alphabet. For example, amino acid Cysteine (C) is encoded as 001⋯0. We then concatenated the numerical representations of all characters of each protein sequence into a single binary feature vector of this spike sequence. In our coronavirus spike protein sequence dataset, after multiple alignments, the length of each spike sequence was found to be 3498. Therefore, the length of each binary vector computed using OHE would be 3498×20= 69,960.

### 3.2. k-mer-Based Frequency Vectors

One of the significant drawbacks of OHE is the high dimensionality of the resulting set of feature vectors. Another problem with OHE is that some (sequential) ordering information on the sequence’s characters (amino acids) is not preserved. An approach that addresses both of these problems is to use sub-strings (also called mers) of length *k*, i.e., *k*-mers. From a sequence, *k*-mers are generated by applying a sliding window of size *k* over the sequences (see Figure 2). Given a sequence of length *N*, the total number of *k*-mers that could be generated is as follows:(1)Totalk-mers=(N−k)+1.

In our experiments to generate *k*-mer-based frequency vectors, we used k=3 (as done in [33,34]).

After generating the *k*-mers, we created a feature vector Φ (a frequency vector), which contains the frequency (count) of each *k*-mer occurring in the sequence [33,34]. Given some sequence σ with alphabet Σ, the length of feature vector Φk(σ) will be |Σ|k. Since we are working with spike protein (amino acid) sequences and taking k=3 in our experiments, the feature vector length we used is 203=8000. This feature vector can be used as input for various ML algorithms. Note that generating sub-strings with a sliding window preserves some (sequential) ordering information on the characters (amino acids) of the (spike protein) sequences, which counters one of the drawbacks of the OHE approach. However, still, to get the frequency count for *k*-mers, we had a high computational cost for bin matching, especially for worst-case searches. Furthermore, the dimensionality of the frequency vectors is still high.

### 3.3. PWM2Vec

Even though the problem of the high dimensionality of the vectors generated in the OHE approach is somewhat mitigated in the *k*-mers approach, the dimensionality of the frequency vectors generated in the *k*-mers approach remains quite high—further improvements can certainly be made. Furthermore, if we can reduce the computational cost for bin matching, that would be a huge improvement in computational cost. To address these problems, we propose PWM2Vec, an approach for generating a fixed-length numerical feature vector based on the concept of the position-weight matrix [71].

While our approach is inspired by the value of the PWM for finding motifs in (e.g., protein) sequences, we used it in a slightly different way in this study. We built a PWM from the *k*-mers of our sequence, and our feature vector was the score of each *k*-mer from this PWM. This allowed us to take advantage of *k*-mers—their ability to capture locality information while also capturing the importance of the position of each amino acid in the sequence (information that is lost in computing the *k*-mer frequency vector). Combining these pieces of information in this way allows us to devise a compact and general feature embedding that can be used in many downstream ML tasks.

Our approach for feature vector generation, PWM2Vec, is summarized in Figure 3. It follows the steps (a–h) explained below. Figure 3a Given the input spike protein sequence shown in Figure 3b, we first extracted the *k*-mers (we used k=9 in the experiments, which was decided using a standard validation set approach [72]). As in Figure 3c, we then generated a position frequency matrix, which contains the frequency count for each character at each position. Note that, in the example, since the (amino acid) sequence is composed of four characters, there are four possible characters at any position. At position 1, for example, in all five *k*-mers, there are two B characters, and so the frequency count of B at position 1 is two. In our experiments, since we had 20 unique amino acids in our spike protein sequence dataset, our PFMs had 20 rows and k=9 columns. Figure 3d Next, we normalized the PFM matrix and created a position probability matrix (PPM) containing the probability of each (amino acid) character at each position. For example, the probability of B in the *k*-mers at position 1 is the following:(2)frequencycounttotalcount=2/5=0.4

It is possible that the frequency (hence probability in the PPM) of a character at a certain position is 0. To avoid 0 values at any position in the matrix while calculating the probability, we added a Laplace estimator (also called pseudocount) to each value in the position probability matrix, as shown in Figure 3e. We used a pseudocount of 0.1 in our experiments [73]. We then computed a position weight matrix (PWM) from the adjusted probability matrix. We made the PWM by computing the log likelihood of each amino acid character *c*, i.e., c∈A,C,⋯,Y, appearing at each position *i* according to
(3)Wc,i=log2p(c,i)p(c){where c∈A,B,C…Z(bases)}

Note that this likelihood was taken under the assumption that the expected frequency of each amino acid is the same (i.e., p(c)=1/|Σ|) because we have 20 amino acids (p(c)=1/20=0.05). Figure 3f shows the computed position weight matrix (PWM). Note that a more nuanced calculation of p(c) specific to amino acid *c* could be performed, based on the number of codons n(c) in *c*, e.g., p(c)=n(c)/61, where 61 is the number of sense codons. Here, n(c)=1 for M and W; 3 for I; 4 for V, P, T, A, and G; 6 for R, L, and S; and 2 for everything else. This, however, carries with it assumption that the four bases coding the nucleotide sequence from which this amino acid sequence originated appear with equal probability. Since we are unsure about such assumptions, and moreover, when testing out this calculation, the results did not change much, we adopted the standard definition of p(c)=1/20. In the future, when we scale this approach to millions of sequences, we will test this empirically to see if there is a separation at this larger scale between these to ways of calculating p(c). This could indicate factors about the sequences, such as GC-enrichment.

After getting the PWM, we used it to compute the absolute scores for each individual *k*-mer generated from the sequence (see Figure 3g for an example). It is the sum of the scores of the bases for the index. The score for *k*-mer (BFDBEDDFF) is shown in Figure 4. The highlighted values in the matrices are summed up to give an absolute score for the *k*-mer, which sums up to 28.28.

Finally, the scores of all the 9-mers are concatenated to get the final feature vector for the given sequence (see Figure 3h for an example). This whole process is repeated for each sequence.

Given a *k*-mer and a PWM, the score for that *k*-mer can be computed as given in Figure 4. The final length of PWM2Vec based feature vector is 3490, which is equal to the number of *k*-mers in each spike sequence.

### 3.4. Feature Selection Method

We use ridge regression (RR) as the feature selection approach, which is commonly used for estimating parameters, thereby addressing collinearity in multiple linear regression model problems [74,75]. This method uses a bias to boost the performance of the model by improving the variance and making the slope more horizontal. This is useful when we need to find out which of the independent attributes are not needed. This gives us the option of removing such columns (attributes) and bringing the slope to zero. The expression for performing ridge regression is the following:(4)min(sumofsquareresiduals+α×slope2)
where α×slope2 is a penalty term. For learning the optimal ridge regression line, we use 5 fold cross validation.

After performing RR in OHE, the total number of selected attributes was 22,322; for the *k*-mers approach it was 7088; and 1616 features were selected for the PWM2Vec-based approach.

## 4. Experimental Setup

This section describes the setup we used for the experiments, followed by the dataset statistics. We also give a visual representation of the data using t-SNE plots. All experiments were conducted using an Intel(R) Xeon(R) CPU E7-4850 v4 @ 2.40 GHz having Windows 10 64 bit OS with 32 GB memory. We implemented our algorithm in Python, and the code is available online for reproducibility (https://github.com/sarwanpasha/PWM2Vec, accessed on 8 March 2022). Our pre-processed data are also available at this link. For classification, we used support vector machine (SVM), naive Bayes (NB), multiple linear regression (MLP), k-nearest neighbors (KNN), random forest (RF), logistic regression (LR), and decision tree (DT). To compute results, we used the 5-fold cross validation approach. We divided the data randomly into training (70%) and testing (30%) sets. For the training data, we used 5-fold cross validation to optimize the parameters (divided the data into 5 equal parts, using 4 parts for training and 1 part for validation) and then tested the performance on the 30% of unseen test data. We repeated this process 5 times (five training–test splits; then we used 5-fold cross validation to tune parameters and compute results on unseen test set) and report the average and standard deviation results.

### 4.1. Evaluation Metrics

To measure the performances of the classifiers, we used average accuracy, precision, recall, weighted, macro F1, and receiver operating characteristic curve “ROC” area under the curve “AUC” (one-vs.-rest approach) using the macro average. We also measured the training time (in seconds) for each classifier.

For clustering, we used the simple *k*-means algorithm and used 3 internal clustering quality metrics, namely, the silhouette coefficient, the Calinski–Harabasz score, and the Davies–Bouldin score to measure the performances of the clusters. We also show the runtimes for *k*-means for different embedding methods.

#### 4.1.1. Silhouette Coefficient

The silhouette coefficient [76] is used for interpretation and validation of consistency within clusters of data. A clustering algorithm having well-defined (comparatively pure) clusters will have a higher silhouette coefficient value. The silhouette coefficient (SC) is computed as follows:(5)SC=y−xmax{y,x}
where *x* is the average distance between a sample and all other points in the data belonging to the same class, and *y* is the average distance between sample *x* and all other data points in the next nearest cluster.

#### 4.1.2. Calinski–Harabasz Score

The Calinski–Harabasz score [77] is used to measure the quality of a clustering algorithm based on the mean between-clusters’ sum of squares. A clustering algorithm with well-defined clusters will have a higher Calinski–Harabasz score. The Calinski–Harabasz score is defined as the ratio of the between-clusters dispersion (the sum of distances squared) mean and the within-clusters dispersion. More formally, given a dataset *D* of size nD that has been clustered into *j* clusters, we used the following expression to compute Calinski–Harabasz (CH) score:(6)CH=tr(Bj)tr(Wj)×(nD−j)(j−1)
where tr(Bj) is the trace of the between cluster dispersion matrix and tr(Wj) is the trace of the within-group dispersion matrix.

#### 4.1.3. Davies–Bouldin Score

The Davies–Bouldin (DB) score [78] of a clustering *C* is defined as follows:(7)DB(C)=1|C|∑i=1|C|maxj≤|C|,j≠iDij
where Dij is the ratio of the “within-to-between cluster distances” of the *i*th and *j*th clusters. For each cluster, we computed the worst case ratio (Dij) of a within-to-between cluster distance between it and any other cluster, and then took the mean. Therefore, by minimizing the DB score, we could make sure that different clusters were separate from each other (a smaller value is better).

### 4.2. Dataset Collection and Statistics

The spike protein sequences of CoVs for all hosts used in this analysis were retrieved (on 21 September 2021) from the NIAD Virus Pathogen Database and Analysis Resource (ViPR) [79] and GISAID [26]. A total of 5568 complete protein sequence were collected (3358 from ViPR and 2210 from GISAID); we later dropped 10 that were not attributable to any host detail. The distribution of the dataset across the different host types (grouped by family) is shown in Table 1, which contains information about the 21 host types that we collected from the annotation of the sequences. We also divided the viral sequences themselves into genera and subgenera to see which category a specific coronavirus belongs to. Figure 5 and Figure 6 contain distributions of viral genera and subgenera, respectively. The multiple sequence alignment (MSA) for the sequence dataset was conducted using the Mafft alignment software with default parameter settings which automatically select the appropriate strategy according to the sequence data size. In our case, the gap opening penalty op was 1.53 and the gap extension penalty ep was 0.123 [80]. Given that our dataset was already sufficiently large and contained a number of unknown or identified amino acids, we were constrained to use the minimum accuracy parameter of Mafft MSA to allow the alignment to complete in a reasonable amount of time. Attempts to set more stringent parameters (op and ep) in order to improve this alignment resulted in runtimes >24 h. Since this is already the case when performing multiple alignments of even  5000 sequences, anything substantially larger is out of reach.

### 4.3. Data Visualization

In order to see if there is any natural (hidden) clustering in the data, we used the t-distributed stochastic neighbor embedding (t-SNE) [81] approach, which maps input sequences to 2D real vectors. The t-SNE plots for different embedding methods are shown in Figure 7, Figure 8 and Figure 9: t-SNE plots for OHE, *k*-mers, and PWM2Vec, respectively. We can observe that although with PWM2Vec, more information is included in lower-dimensional feature vectors, the proposed embedding approach was able to preserve the structure of data similarly to OHE and *k*-mers.

## 5. Results and Discussion

In this section, we present our results for PWM2Vec and compare its performance with the baseline one-hot embedding (OHE) and the more recent *k*-mer-based embedding approach, which has shown to be an improvement over OHE [33,34]. For classification, we also show the results for the feature selection method (ridge regression) for all embedding approaches. We also show the runtimes with different numbers of sequences for the best-performing classification algorithm. Finally, we show some statistical analysis on the data and on the feature vectors computed using different approaches.

### 5.1. Classification Results

Table 2 shows the average results for various embedding methods with various classification methods without performing any feature selection on the feature vectors. The standard deviations for 5-fold cross validation are shown in Table 3. We can see that RF with PWM2Vec is consistently performing better than other embedding methods (in some cases, the performance gain for PWM2Vec is on the third significant digit). Furthermore, the NB classifier with PWM2Vec was much better than other approaches in terms of training runtime. This behavior shows that PWM2Vec is not only better in terms of predictive performance but is also better in terms of runtime.

Table 4 contains averages of the classification results after applying ridge regression to different feature embedding approaches. The standard deviation values are shown in Table 5. We can again see that RF with PWM2Vec outperformed other embedding methods for the majority of the metrics, and NB with PWM2Vec was the best in terms of runtime.

#### Effect on Runtime

To evaluate the effect on runtime of the sequences, we used the best performing classifier (random forest) and used different embedding methods to perform classifications with increasing numbers of sequences. Figure 10 shows the runtimes for (a) OHE vs. PWM2Vec and (b) *k*-mers vs. PWM2Vec. In both cases, we can see that PWM2Vec is better in terms of runtime as we increase the number of sequences (on the x-axis).

### 5.2. Clustering Results

For clustering, we used the same feature embeddings that we used for the classification task. To get the optimal number of clusters, we used the elbow method [35]. This method, for different numbers of clusters (ranging from 2 to 30), performed clustering to see the trade-off between the runtime and the sum of squared error (distortion score). The optimal number of clusters selected was nine (see Figure 11).

For the purposes of clustering, we used the simple *k*-means algorithm. The results of clustering on different embedding methods are shown in Table 6. We can see that PWM2Vec was better in terms of silhouette coefficient and runtime, whereas the *k*-mer-based approach was better in terms of Calinski–Harabasz score and Davies–Bouldin score.

### 5.3. Statistical Analysis

To measure the importance of amino acids corresponding to the class label, we used the information gain (IG). The IG is defined as follows:(8)IG(Class,position)=H(Class)−H(Class|position)
(9)H=∑i∈Class−pilogpi
where *H* is the entropy, and pi is the probability of class *i*. Figure 12 shows the IG values for different amino acids corresponding to the class labels (hosts). We can see that some amino acids have higher IG values, which means that they play an important role in the prediction of hosts. Here, we can conclude that many amino acids contribute to the host specification, and less to SARS-CoV-2 variant specification [33], which was expected since the genomic variability within the family Coronaviridae should be much higher. The IG values for all amino acids are also available online for further analysis (https://github.com/sarwanpasha/PWM2Vec/tree/main/IG%20values, accessed on 8 March 2022).

When inspecting the IG values and how they are distributed more closely, in the range from 0 to 1.30 with a mean (μ) of 0.373 and a standard deviation (σ) of 0.438, we can see that 29 positions are more than 2σ from this mean. In inspecting how these 29 high IG positions are distributed along the length of the amino acid sequence, we see that they range from 734 to 3443, with a median position 1458, and μ±σ=1610±747. Since the average Coronaviridae spike sequence of our dataset has length ≈1500, this means that these high IG values tend to concentrate in the latter half of the spike sequence, which corresponds to the S2 subunit of the spike protein. In the case of SARS-CoV-2, the most studied of the Coronaviridae, the S2 subunit is composed of five domains, namely, the fusion peptide (FP), two heptad-repeat regions (HR1 and HR2), the transmembrane domain (TM), and the cytoplasmic tail (CT), and they are responsible for mediating viral cell membrane fusion and entry [20]. Each unit of this S2 subunit has been shown to play a key role in spike protein fusion activities [20,21]. Since the typical Coronaviridae sequence has a similar structure (see, e.g., https://www.uniprot.org/uniprot/P11224, accessed on 8 March 2022), it is believed that the typical S2 subunit carries out similar functions. The fact that these S2 subunits play important roles could indicate why such a range of positions have high IG values. A more complete study of this in terms of details of peptide structure would be an interesting future study.

Since information gain does not give us the negative (or opposite) contribution of an attribute (feature) corresponding to the class label (host names), we used other statistical measures, such as Spearman correlation [82], to further evaluate the contributions of features in the PWM2Vec-based feature vector. The Spearman correlation is computed using the following expression:(10)ρ=1−6∑di2n(n2−1)
where ρ is the Spearman’s rank correlation coefficient, di is the difference between the two ranks of each observation, and *n* is the total number of observations.

The Spearman correlation and corresponding p-values for PWM2Vec are given in Figure 13. We can observe that most of the features contribute towards the prediction of different hosts. Table 7 contains the comparisons of correlation values computed using Spearman correlation for the different embedding methods. Here, we can observe that for a lower dimensional and more compact approach for feature embedding (PWM2Vec), the fraction of features having correlation values greater than the threshold (i.e., 0.3 and −0.3) is greater than that fraction generated by OHE, and is comparable with those given for *k*-mers (sometimes better also). This behavior shows that by using PWM2Vec, we were able to preserve more information in a smaller feature vector and improve the runtimes of underlying ML algorithms while giving better (sometimes comparable) predictive performance.

## 6. Conclusions

We proposed an approach called PWM2Vec to generate feature vector representations for the host preferences of different coronaviruses using spike sequences only, which can be used as input for different machine learning algorithms, such as classification and clustering. We show that our approach is not only efficient for generating feature vectors as compared to the popular method based on *k*-mers, but has comparable prediction accuracies and a much shorter training runtime. This behavior was also observed after applying the feature selection algorithm. We also provided some statistical analysis on the data and feature vectors to show the importance of attributes towards the prediction of class labels (hosts). This statistical analysis provided validation, from an independent point of view (in terms of the fraction of features statistically correlated to the label), of the compactness of our PWM2Vec embedding, compared to the baselines. In the future, we will focus on collecting more data to evaluate the scalability of PWM2Vec. Using unsupervised methods for dimensionality reduction is another future extension of this work. We would also like to use deep learning models such as LSTM and GRU for classification purposes in the future. The application of this to larger families of viruses could also be another interesting future direction. Using information gain and correlations (e.g., Spearman correlation) to study the structure of spike protein is another interesting future direction.

## Figures and Tables

**Figure 1 biology-11-00418-f001:**
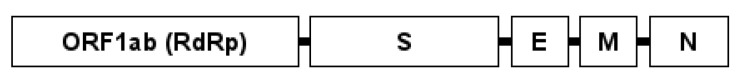
The genome of a coronavirus ranges from 26 to 32 kb in length [5], each of them coding for two non-structural and four structural proteins. The non-structural proteins are coded in ORF1ab, which contains the RNA-dependent RNA polymerase gene (RdRp). The structural genes include spike (S), envelope (E), membrane (M), and nucleocapsid (N). The S gene region encodes the spike protein, which is responsible for attaching the virus to receptors on the host cell membrane.

**Figure 2 biology-11-00418-f002:**
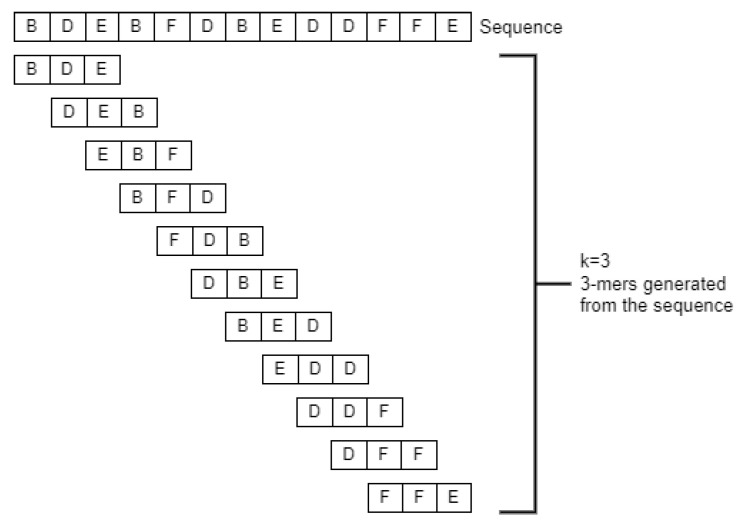
Generating 3-mers (k=3) from a spike protein sequence using a sliding window. Since this sequence has length 13, the total number of *k*-mers generated is 13 − 3 + 1 = 11 (Equation (Equation 1)).

**Figure 3 biology-11-00418-f003:**
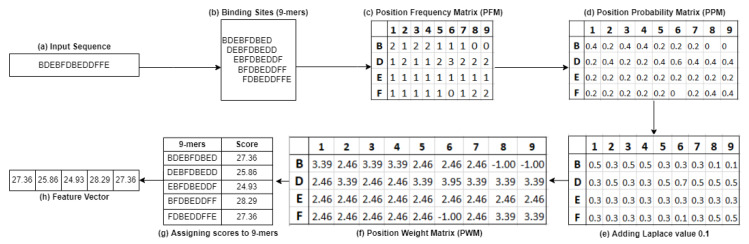
Building a feature vector for classification models by computing the position weight matrix (PWM) from the *k*-mers of a sequence.

**Figure 4 biology-11-00418-f004:**
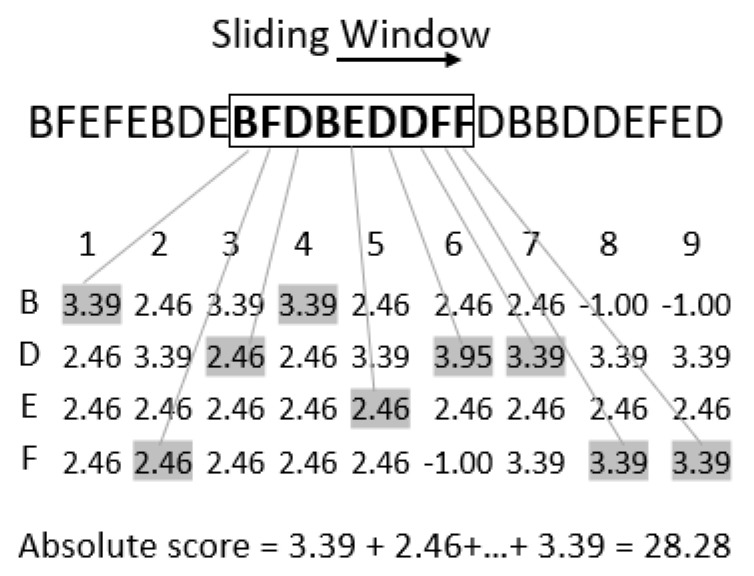
Computing the score from the PWM using a sliding window on a sequence for a 9-m.

**Figure 5 biology-11-00418-f005:**
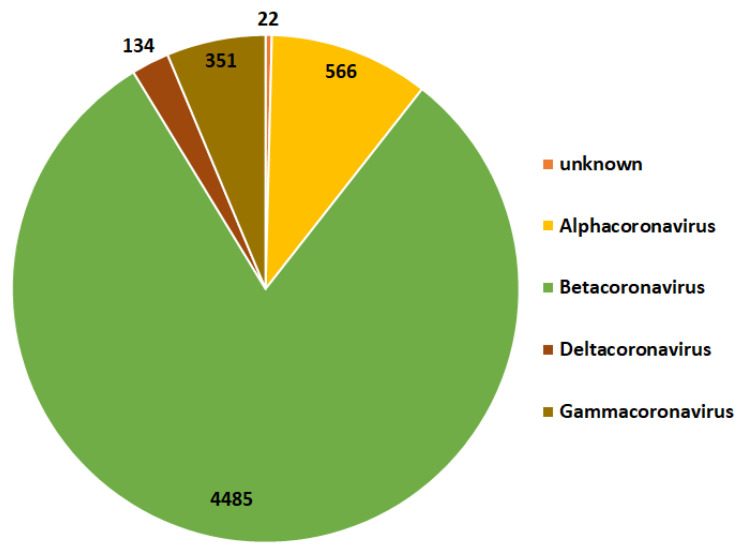
Pie chart for the distribution of different genera in the dataset.

**Figure 6 biology-11-00418-f006:**
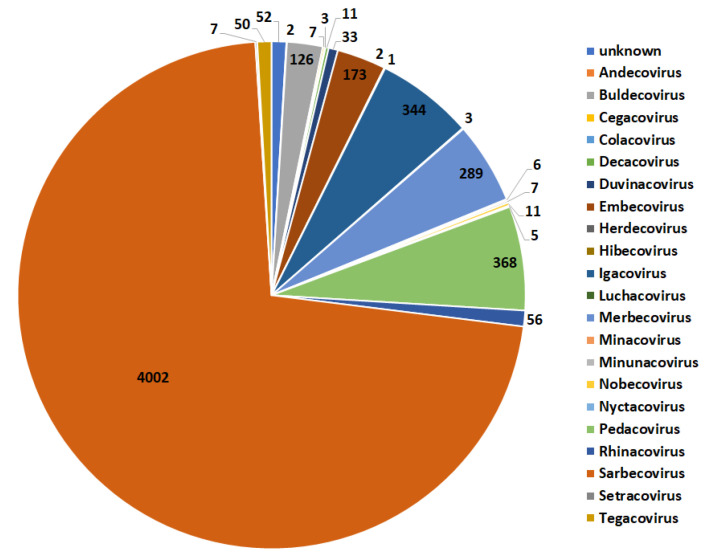
Pie chart for the distribution of different subgenera in the dataset.

**Figure 7 biology-11-00418-f007:**
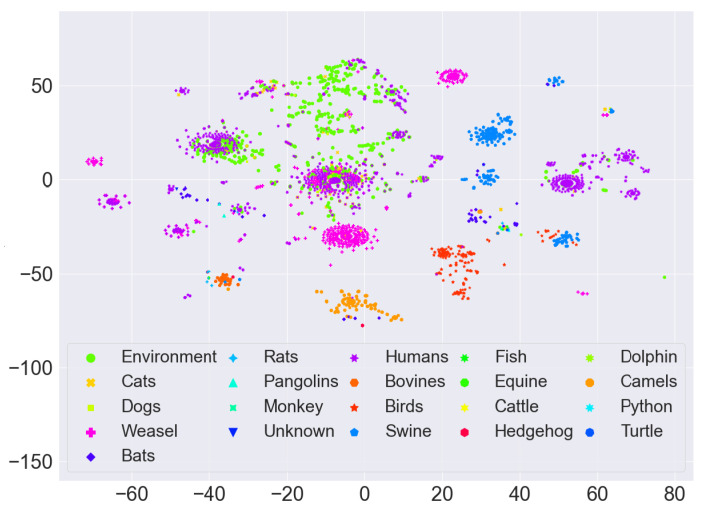
t-SNE plots for one-hot encoding.

**Figure 8 biology-11-00418-f008:**
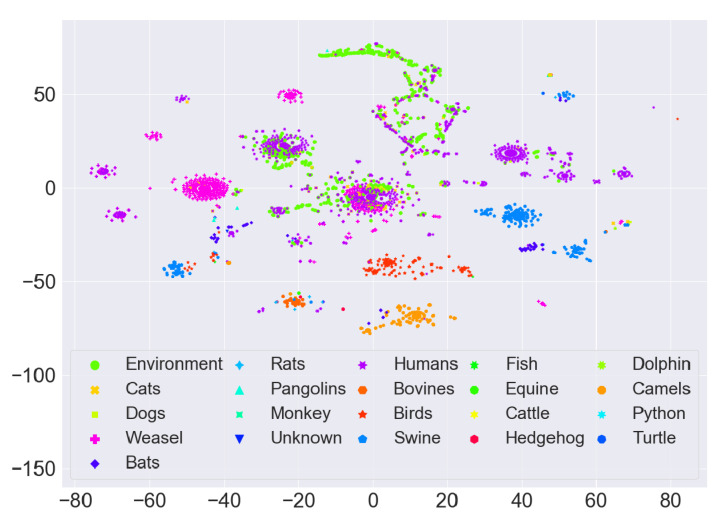
t-SNE plots for *k*-mers.

**Figure 9 biology-11-00418-f009:**
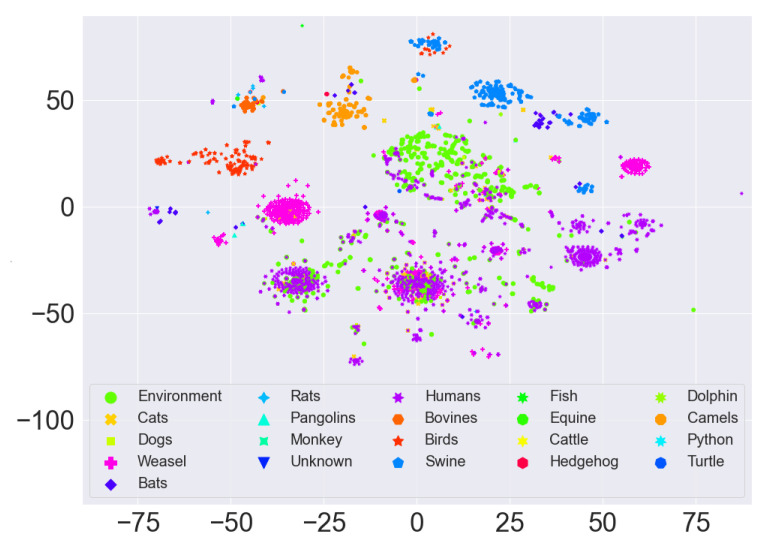
t-SNE plots for PWM2Vec.

**Figure 10 biology-11-00418-f010:**
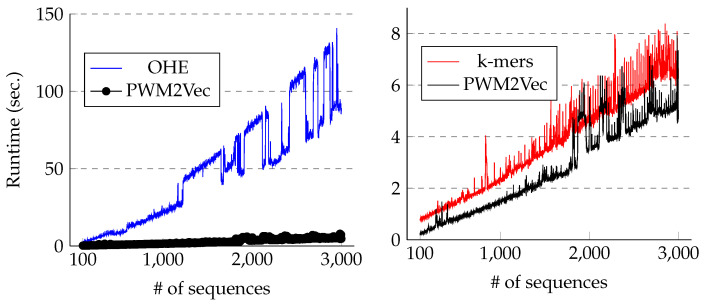
Runtime comparison for different embedding methods with increasing numbers of sequences using the random forest classifier (best performing classifier). The figure is best seen in color.

**Figure 11 biology-11-00418-f011:**
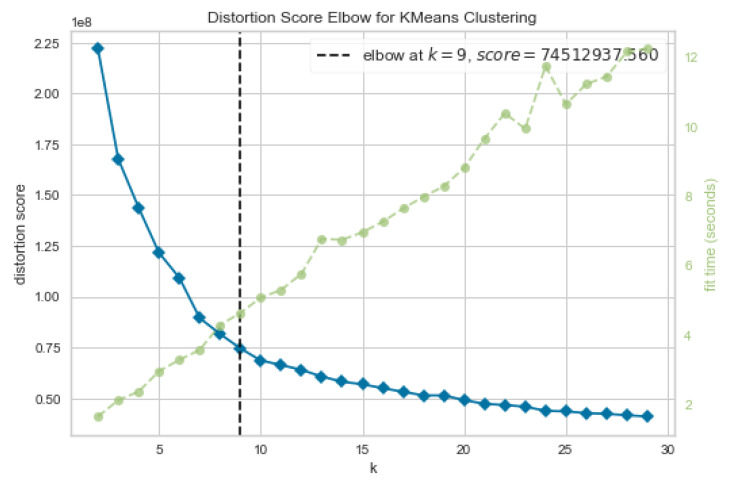
The elbow method for optimal number of clusters.

**Figure 12 biology-11-00418-f012:**
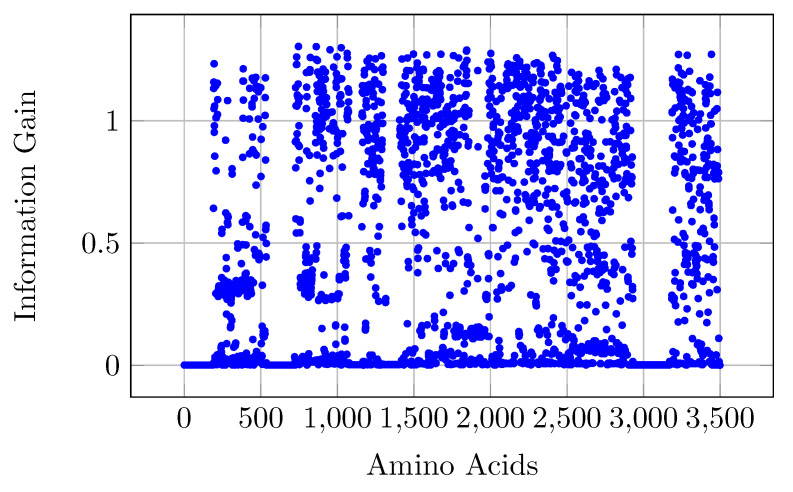
Information gain for each amino acid position with respect to hosts.

**Figure 13 biology-11-00418-f013:**
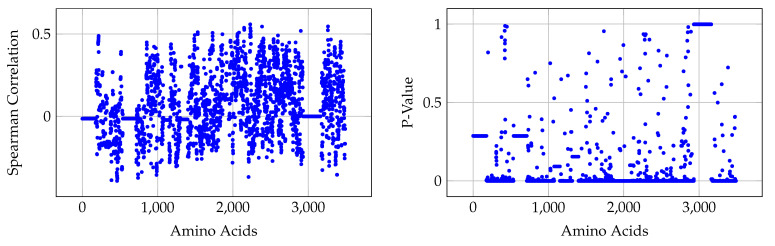
Spearman correlation for PWM2Vec.

**Table 1 biology-11-00418-t001:** Dataset statistics for 5558 coronavirus hosts.

Host Name	# of Sequences	Host Name	# of Sequences
Humans	1813	Rats	26
Environment	1034	Pangolins	21
Weasel	994	Hedgehog	15
Swine	558	Dolphin	7
Birds	374	Equine	5
Camels	297	Fish	2
Bats	153	Unknown	2
Cats	123	Python	2
Bovines	88	Monkey	2
Dogs	40	Cattle	1
Turtle	1		

**Table 2 biology-11-00418-t002:** Performance comparison (average results for 5-fold cross validation) for different embedding methods and different classifiers without using any feature selection approach. Best values are shown in bold.

		Acc.	Prec.	Recall	F1 (Weig.)	F1 (Macro)	ROC AUC	Train Time (S)
	SVM	0.81	0.82	0.81	0.81	0.69	0.82	389.128
	NB	0.68	0.81	0.68	0.66	0.65	0.80	56.741
	MLP	0.76	0.75	0.76	0.74	0.43	0.70	390.289
OHE	KNN	0.79	0.78	0.79	0.78	0.54	0.77	16.211
	RF	0.83	0.83	0.82	0.82	0.66	0.82	151.911
	LR	0.82	0.83	0.83	0.82	0.70	0.83	48.786
	DT	0.82	0.83	0.82	0.81	0.63	0.80	21.581
	SVM	0.80	0.81	0.80	0.81	0.64	0.82	52.384
	NB	0.64	0.76	0.66	0.65	0.47	0.73	9.031
	MLP	0.81	0.82	0.81	0.81	0.52	0.77	44.982
*k*-mers	KNN	0.81	0.80	0.81	0.79	0.55	0.75	2.917
	RF	0.82	0.83	0.83	0.81	0.63	0.81	17.252
	LR	0.82	0.84	0.81	0.82	0.68	0.82	48.826
	DT	0.81	0.82	0.81	0.80	0.64	0.81	4.096
	SVM	0.80	0.81	0.80	0.81	0.71	**0.85**	40.55
	NB	0.46	0.70	0.46	0.40	0.47	0.76	**1.56**
	MLP	0.80	0.81	0.81	0.79	0.57	0.78	17.28
PWM2Vec	KNN	0.82	0.81	0.82	0.81	0.58	0.79	2.86
	RF	**0.85**	**0.85**	**0.85**	**0.84**	**0.72**	0.84	5.44
	LR	0.82	0.82	0.82	0.82	0.71	0.84	43.35
	DT	0.81	0.81	0.82	0.81	0.66	0.83	3.46

**Table 3 biology-11-00418-t003:** Performance comparison (standard deviation results for 5-fold cross validation) for different embedding methods and different classifiers without using any feature selection approach.

		Acc.	Prec.	Recall	F1 (Weig.)	F1 (Macro)	ROC AUC
	SVM	0.0093	0.0105	0.0097	0.0103	0.0389	0.0292
	NB	0.0156	0.0144	0.0155	0.0177	0.0255	0.0202
	MLP	0.0024	0.0064	0.0026	0.0042	0.0493	0.0300
OHE	KNN	0.0121	0.0115	0.0126	0.0133	0.0558	0.0270
	RF	0.0195	0.0095	0.0115	0.0117	0.0582	0.0262
	LR	0.0083	0.0048	0.0084	0.0093	0.0307	0.0236
	DT	0.0179	0.0162	0.0176	0.0185	0.0536	0.0298
	SVM	0.0095	0.0101	0.0091	0.0102	0.0375	0.0235
	NB	0.0132	0.0164	0.0125	0.0126	0.0263	0.0245
	MLP	0.0012	0.0063	0.0029	0.0035	0.0465	0.0339
*k*-mers	KNN	0.0118	0.0191	0.0135	0.0173	0.0594	0.0227
	RF	0.0135	0.0063	0.0116	0.0193	0.0541	0.0235
	LR	0.0063	0.0025	0.0043	0.0016	0.0325	0.0269
	DT	0.0132	0.0135	0.0165	0.0142	0.0519	0.0148
	SVM	0.0025	0.0165	0.0043	0.0125	0.0343	0.0232
	NB	0.0143	0.0142	0.0156	0.0175	0.0251	0.0204
	MLP	0.0056	0.0054	0.0027	0.0075	0.0475	0.0303
PWM2Vec	KNN	0.0125	0.0174	0.0193	0.0124	0.0568	0.0210
	RF	0.0185	0.0035	0.0165	0.0143	0.0565	0.0296
	LR	0.0043	0.0057	0.0093	0.0015	0.0357	0.0253
	DT	0.0136	0.0193	0.0156	0.0165	0.0542	0.0235

**Table 4 biology-11-00418-t004:** Performance comparison (average results for 5-fold cross validation) for different embedding methods and different classifiers using ridge regression as the feature selection approach. Best values are shown in bold.

		Acc.	Prec.	Recall	F1 (Weig.)	F1 (Macro)	ROC AUC	Train Time (s)
	SVM	0.83	0.83	0.83	0.82	0.67	0.81	63.992
	NB	0.63	0.75	0.63	0.61	0.53	0.77	9.436
	MLP	0.82	0.82	0.82	0.80	0.51	0.75	64.636
OHE	KNN	0.78	0.78	0.78	0.78	0.61	0.81	2.730
	RF	0.83	0.83	0.83	0.82	0.59	0.81	22.423
	LR	0.83	0.82	0.83	0.83	0.65	0.84	26.094
	DT	0.83	0.83	0.83	0.82	0.60	0.83	6.316
	SVM	0.81	0.81	0.81	0.81	0.73	0.87	30.877
	NB	0.67	0.78	0.67	0.67	0.64	0.83	4.012
	MLP	0.83	0.83	0.83	0.83	0.58	0.81	26.280
*k*-mers	KNN	0.80	0.80	0.80	0.80	0.69	0.83	1.601
	RF	0.82	0.82	0.81	0.82	0.73	0.87	6.786
	LR	0.82	0.82	0.81	0.82	0.79	0.88	39.501
	DT	0.83	0.83	0.83	0.83	0.70	0.88	2.429
	SVM	0.78	0.79	0.78	0.78	0.75	0.89	24.53
	NB	0.41	0.64	0.41	0.40	0.38	0.68	0.94
	MLP	0.81	0.81	0.81	0.80	0.67	0.82	9.85
PWM2Vec	KNN	0.80	0.80	0.80	0.79	0.62	0.80	1.55
	RF	**0.84**	**0.84**	**0.84**	**0.85**	**0.80**	0.86	5.06
	LR	0.80	0.81	0.80	0.80	0.66	**0.90**	21.76
	DT	0.80	0.80	0.80	0.80	0.64	0.82	2.00

**Table 5 biology-11-00418-t005:** Performance comparison (standard deviation results for 5-fold cross validation) for different embedding methods and different classifiers using ridge regression as the feature selection approach.

		Acc.	Prec.	Recall	F1 (Weig.)	F1 (Macro)	ROC AUC
	SVM	0.0099	0.0108	0.0097	0.0091	0.0481	0.0232
	NB	0.0235	0.0232	0.0231	0.0138	0.0431	0.0270
	MLP	0.0147	0.0146	0.0143	0.0154	0.0544	0.0252
OHE	KNN	0.0122	0.0132	0.0121	0.0124	0.0567	0.0267
	RF	0.0090	0.0108	0.0098	0.0099	0.0690	0.0242
	LR	0.0075	0.0085	0.0073	0.0084	0.0432	0.0258
	DT	0.0118	0.0135	0.0113	0.0128	0.0538	0.0278
	SVM	0.0092	0.0106	0.0092	0.0087	0.0492	0.0237
	NB	0.0230	0.0245	0.0231	0.0135	0.0439	0.0277
	MLP	0.0137	0.0133	0.0136	0.0141	0.0552	0.0259
*k*-mers	KNN	0.0130	0.0141	0.0131	0.0130	0.0614	0.0289
	RF	0.0096	0.0117	0.0093	0.0106	0.0684	0.0249
	LR	0.0075	0.0083	0.0079	0.0085	0.0438	0.0242
	DT	0.0128	0.0147	0.0127	0.0139	0.0582	0.0274
	SVM	0.0096	0.0096	0.0097	0.0094	0.0540	0.0261
	NB	0.0205	0.0256	0.0205	0.0147	0.0382	0.0295
	MLP	0.0150	0.0108	0.0156	0.0152	0.0656	0.0313
PWM2Vec	KNN	0.0145	0.0164	0.0144	0.0151	0.0666	0.0312
	RF	0.0112	0.0131	0.0113	0.0121	0.0773	0.0288
	LR	0.0081	0.0092	0.0087	0.0093	0.0403	0.0282
	DT	0.0149	0.0169	0.0150	0.0162	0.0565	0.0311

**Table 6 biology-11-00418-t006:** Internal clustering quality metrics for *k*-means. Best values are show in bold.

	Evaluation Metrics	
Methods	Silhouette Coefficient	Calinski–Harabasz Score	Davies–Bouldin Score	Runtime (s)
OHE	0.631	2210.343	1.354	177.54
*k*-mers	0.735	**14,296.17**	**0.534**	36.57
PWM2Vec	**0.750**	2563.547	1.314	**23.67**

**Table 7 biology-11-00418-t007:** Correlation values for different embedding approaches computed using Spearman correlation. We show the count (No.) and fraction (frac.) of feature values greater than or less than the threshold (0.3 or −0.3). The fractions were computed by taking the size of the embedding as the denominator (69,900 for OHE, 8000 for *k*-mers, and 3490 for PWM2Vec).

	Spearman Correlation
	>0.3	<−0.3
Methods	No.	Frac.	No.	Frac.
OHE	664	0.007	971	0.011
*k*-mers	557	0.040	705	0.050
PWM2Vec	419	0.120	33	0.009

## Data Availability

Data for this study were obtained from the public databases Virus Pathogen Database and Analysis Resource (ViPR) (https://www.viprbrc.org/) and the Global Initiative on Sharing All Influenza Data (GISAID) (https://www.gisaid.org/) on 21 September 2021. Codes and pipelines for reproducing the results can be found at https://github.com/sarwanpasha/PWM2Vec (accessed on 8 March 2022).

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
