# Peer review of "PWM2Vec: An Efficient Embedding Approach for Viral Host Specification from Coronavirus Spike Sequences"

_biology, 2022, doi:10.3390/biology11030418_

Round 1
Reviewer 1 Report
The manuscript “PWM2Vec: An efficient embedding approach for viral host specification from coronavirus spike sequences” by Ali and colleagues describes a classifier using methods inspired by standard position-specific specific weight matrices that can be used to identify hosts of coronaviruses based on spike proteins. I am intrigued by the method and think it has potential, but I see one fundamental issue and some minor issues.
The fundamental issue I see is the nature of the amino acid alphabet. The authors state that there are 24 amino acid (alphabet = ABCDEFGHIJKLMNPQRSTVWXYZ) when there are actually 20 major proteinogenic amino acids. Four of the symbols that the authors include are ambiguity codes (B=N or D, J=I or L, X=any amino acid, Z=Q or E). Moreover, the ambiguity symbols were largely chosen due to the peculiarities of older sequencing technologies (e.g., B and Z reflect the fact that standard Edman degradation cannot distinguish between N/D and Q/E; see. Airoldi and Doonan 1975). However, there is no reason why amino acid sequences that reflect the translation of DNA sequences would produce B, J, or Z (of course, in an organism with a diploid genome a coding region could be heterozygous for N/D, I/L, or Q/E, but there are myriad other heterozygosities possible, most of which would be written as X). This means that “X” would be the only ambiguity symbol that is commonly used for amino acid sequences derived from DNA sequencing data and that the alphabet seems inappropriate. This raises a few questions in my mind:
- How often did the ambiguity symbols appear in their sequences?
- If they did appear, what did they mean? Since the viruses are not diploids that cannot be heterozygous sites.
- Does the use of an inappropriate alphabet matter? I can see an argument that they would not. If we view amino acids as equiprobable (cf. lines 273-274 in the manuscript) then the biological 20-letter alphabet yields p(c)=1/20=0.05 when the authors have p(c)=1/24=0.041. Perhaps the difference between 0.05 and 0.041 is so trivial as to not matter. I wrestled with this issue quite a bit and I’d like the authors to address it carefully.
- The issue of calculating p(c) as the reciprocal of the number of letters in the alphabet is itself somewhat troubling. After all, the number of codons for each amino acid varies from 1 to 6. I could see a strong argument for using p(c)=n(c)/61, where n(c) is the number of codons for each amino acid (i.e., 1 for MW, 2 for CFYHQNKDE, 3 for I, 4 for VPTAG, and 6 for RL) and 61 is the number of sense codons. Of course, that calculation itself assumes that the four bases in the coding strand are equiprobable (if GC is >50% one would expect increased GARP and decreased FYMINK, the opposite if coding-strand GC is 50%).
My bigger concern about this really comes down to the impact of p(c) on their calculations. I think the authors should really examine the sensitivity to p(c)=0.05 for all c and p(c)=n(c)/61.
Another big concern is the fact that the appearance of clustering in t-SNE plots can be misleading (see “web information on t-SNE plots below). It would be really nice to create random sequences with the same amino acid composition as the spike proteins and see how they appear to cluster using the same t-SNE parameters. Ideally, there would be little or no clustering in the t-SNE plots for random spike proteins. Of course, the clustering could be recovering phylogeny. This is another thing the authors should consider. Have the authors generated a spike protein phylogeny using a program like IQ-TREE? Or even parsimony? (see Thornlow et al. 2021).
Beyond the methodological and philosophical issues of t-SNE, some of the symbols in the t-SNE plots are difficult to read. For example, I spent a lot of time staring at Figure 10 and could not differentiate between weasel, unknown, and turtle. Since there are three major clusters of red plus some points elsewhere it wasn’t clear if weasel was just found in multiple clusters or if the different red clusters were different hosts.
It would be nice to look at the information gain and correlations (Figures 13-15) in light of spike protein structure. There are clear patterns and I suspect high information gain will be solvent-exposed extracellular residues. This should not be that hard to examine given the structural data available for coronavirus spike proteins. This could even be done adequately by using of solvent accessibility prediction methods (like ACCpro; Pollastri et al. 2002; Cheng et al. 2005) and combining the classification of residues as solvent exposed versus buried in light of the annotations of extracellular, transmembrane, and intracellular boundaries in uniprot (for an example see the uniprot record https://www.uniprot.org/uniprot/P11224).
I am also puzzled by the use of both parametric (Pearson) and non-parametric (Spearman) correlations. I pick one or the other – in general, I prefer Spearman unless I am certain that the assumptions of Pearson’s correlation are met. On the other hand, Pearson correlations are relatively robust so it may not matter. However, I don’t see a reason to do both.
There are some other communication issues, like the general use of common names for hosts but the occasional use of more scientific names (e.g., Table 1, which includes mostly common names but also includes “Canis” rather than “Dog”). Also, Table 1 includes both singular and plural, sometimes using inappropriate plurals (“Swine” is both singular and plural, so “Swines” is incorrect). I think the authors need to be consistent and I suggest they use common names and singular. Likewise, the authors should not use “avian” when “bird” or “birds” is more appropriate (e.g., lines 38-39). “Avian” should be used as an adjective, “bird” or “birds” as a noun.
The first sentence in the abstract: “The origin of SARS-CoV-2 in humans, which led to the COVID-19 pandemic, is still unknown and is an important open question.” seems to be a red herring. The origin of SARS-CoV-2 is a phylogenetic question; the authors are trying to look at likely host specificity.
Finally, the map of the “typical” coronavirus genome in Figure 1 seems off. The ORF1ab polyprotein is a single region with a programmed frameshift. This will lead to the generation of a partial polyprotein (just ORF1a) or, if a ribosomal frameshift occurs, a complete polyprotein (ORF1ab). As drawn the map implies two copies of ORF1ab, which is not correct.
Overall, I feel positive about this manuscript. Parts of it read like a primer on the relevant methods, but I kind of like this. I do feel that the issue of using a 20-state rather than 24-state alphabet is important, just because the 20-state alphabet is the biologically relevant one. I think this method could be generally useful beyond coronaviruses so I hope the authors will consider these issues carefully. I hope these comments are helpful.
References for this review:
Airoldi, L. P. D. S., & Doonan, S. (1975). A method of distinguishing between aspartic acid and asparagine and between glutamic acid and glutamine during sequence analysis by the dansyl-Edman procedure. FEBS letters, 50(2), 155-158.
- Cheng, A. Randall, M. Sweredoski, P. Baldi, SCRATCH: a Protein Structure and Structural Feature Prediction Server, Nucleic Acids Research, vol. 33 (web server issue), w72-76, 2005.
G.Pollastri, P.Baldi, P.Fariselli, R.Casadio, "Prediction of Coordination Number and Relative Solvent Accessibility in Proteins", Proteins, 47, 142-153, 2002.
Thornlow, Bryan, Cheng Ye, Nicola De Maio, Jakob McBroome, Angie S. Hinrichs, Robert Lanfear, Yatish Turakhia, Russell Corbett-Detig. Online Phylogenetics using Parsimony Produces Slightly Better Trees and is Dramatically More Efficient for Large SARS-CoV-2 Phylogenies than de novo and Maximum-Likelihood Approaches. bioRxiv 2021.12.02.471004; doi: https://doi.org/10.1101/2021.12.02.471004
Web information on t-SNE plots: https://stats.stackexchange.com/questions/263539/clustering-on-the-output-of-t-sne/264647#264647
Reviewer 2 Report
The paper is well written can be accpted but cnsider these points
Reviewer comments:
The paper is too medium size
Introduction can be elaborated
According author guidelines formatting is required
Authors used feature embedding based 10 on the well-known position-weight matrix (PWM), claimed they used first .
Manuscript available in this website https://arxiv.org/pdf/2201.02273.pdf(editors chek)
Round 2
Reviewer 1 Report
I feel the author's have addressed my concerns. I did notice one mistake that should be corrected at the stage of copy editing ("Spearman" is not capitalized in line 466), but I feel the scientific issues have been addressed.
Author Response
We have done a pass of the paper for English grammar, spelling, style and consistency. We thank the reviewer for this comment